



# Experimental results of wake steering using fixed angles

Paul Fleming[1], Michael Sinner[1], Tom Young[2], Marine Lannic[2], Jennifer King[1], Eric Simley[1], and Bart Doekemeijer[1]

[1]National Wind Technology Center, National Renewable Energy Laboratory, Golden, CO, 80401, USA
[2]RES Group, Beaufort Court, Egg Farm Lane, Kings Langley, Hertfordshire, WD4 8LR, UK

**Correspondence:** Paul Fleming (paul.fleming@nrel.gov)

**Abstract.**

In this article, the authors present a test of wake steering at a commercial wind farm. A single fixed yaw offset, rather than an optimized offset schedule, is alternately applied to an upstream wind turbine and the effect on downstream turbines is analyzed. This experimental design allows for comparison with engineering wake models independent of the controller's ability to track a varying offset and correctly measure wind direction. Additionally, by applying the same offset in beneficial and detrimental conditions, we are able to collect important data for assessing second-order wake model predictions. Results of the article from collected data show good agreement with the FLOw Redirection and Induction in Steady State (FLORIS) engineering model and offer support for the asymmetry of wake steering predicted by newer models, such as the Gauss-curl hybrid model.

## Copyright Statement

## 1   Introduction

Wake steering is a form of wind farm control in which intentional yaw misalignments are applied to upstream wind turbines to change their wakes to benefit downstream turbines.Wagenaar et al. (2012) Research has demonstrated that by implementing well-designed control strategies, it is possible to increase the combined power production of a given set of wind turbines.

Engineering, or control-oriented, models of wakes and wake steering are critical to the design of successful control strategies. In order to be used in the optimizations that design controllers, or perhaps better in the online control system itself, it is necessary that the models be computationally efficient. They are therefore typically analytical, or statistical, in nature.



The FLOw Redirection and Induction in Steady State (FLORIS) software framework (Laboratory (2019)) is one such engi-
neering tool, and it includes several engineering models of wake and wake steering, as well as the tools used in the design and
analysis of wind farm control strategies. The original FLORIS model, detailed in Gebraad et al. (2016), combined a multizone
adaptation of the Jensen/Park model (Jensen (1984)) with a model of wake steering as deflection, as described in Jiménez et al.
(2010).

The current underlying wake model used in FLORIS-based design and analysis is the Gaussian wake model described in
Bastankhah and Porté-Agel (2014); Niayifar and Porté-Agel (2015) and the model of deflection provided in Bastankhah and
Porté-Agel (2016). However, this model is modified by an analytic approximation of the curl model Martínez-Tossas et al.
(2019) to adjust the deflection of nonsteered wakes by interaction with steered wakes (secondary steering), as well as an
effect called "yaw-added recovery," which adjusts the velocity recovery of steered wakes. This model is called the Gauss curl
hybrid (GCH) and is presented in King et al. (2020). Since its incorporation into FLORIS, it is the standard tool used in
design, optimization, and analysis of wind farm control, because of its ability to predict the second-order effects of "secondary
steering" and "yaw-added recovery," in addition to wake deflection.

In order to achieve optimal results, it is necessary to validate the predictions of engineering models like FLORIS. Original
validation experiments of FLORIS were compared to the large-eddy, Simulator fOr Wind Farm Applications (SOWFA)(Churchfield
et al. (2014).) Wake steering is compared to SOWFA in Fleming et al. (2014), Fleming et al. (2015), Fleming et al. (2018) and
King et al. (2020). Validation of the engineering models with respect to higher-fidelity simulations is still ongoing.

There is also growing literature on wind tunnel investigations used to assess engineering models of wake steering. These
include two-turbine (Adaramola and Krogstad (2011), Bartl et al. (2018), Schottler et al. (2016), Zong and Porté-Agel (2020)),
three-turbine (Campagnolo et al. (2016a), Campagnolo et al. (2016b), Park et al. (2016)) and five-turbine (Bastankhah and
Porté-Agel (2019)) studies. These experiments allow for controlled and detailed experimentation and examination of wake
steering.

There are further experiments of wake steering completed at full scale. In one experiment, a 1.5-MW wind turbine had a
rear-facing lidar installed scanning the wake while different fixed yaw offsets were applied in Fleming et al. (2017a). The
results were then compared against several models within FLORIS in Annoni et al. (2018).

Finally, there are tests of wake steering at commercial-scale sites. These include tests made using fixed offsets over a fixed
sector of wind directions (Ahmad et al. (2019), Howland et al. (2019)) (in both cases 40-deg sectors are used), and offsets
applied using an optimal schedule based on wind directions (Fleming et al. (2017b), Fleming et al. (2019), Fleming et al.
(2020), and Doekemeijer et al. (2021)).

In this article, we review the results of a new wake-steering experiment. The campaign is different from previous campaigns
conducted by the NREL in that fixed offsets (as opposed to a look-up table of optimal offsets per wind direction) are applied for
all directions where the wind turbine's wake impacts another turbine. This sort of test has several useful attributes. First, by not
varying the offset amount, wake-steering effects can be observed at full scale isolated from issues of tracking a changing offset
amount. In Fleming et al. (2020), for example, it is posited that the main gap in achieved versus modeled wake-steering results
is attributable to this tracking/controller problem. This experiment offers a test of that supposition. A second advantage is that





because the same offset is applied whether the wake would be steered away from a downstream wind turbine (normal wake
steering) or toward a downstream wind turbine (what might be called "wrong-way steering"), we can examine the asymmetry
of wake steering. This provides an important test of the "yaw-added recovery" component of the GCH model of wake steering.
Results show agreement between model predictions and measured results.

## 2    Test Site Overview and Pretest Period

We conducted the experiment on an 11-turbine wind farm illustrated in Fig. 1. In the test, we applied offsets to the yaw
controller of a single wind turbine. This turbine, T5, is referred to as the controlled turbine and is indicated in Fig. 1. A second
turbine, T4, is designated as the reference and is in the freestream for the wind directions of interest. Finally, turbines T6
and T10 are the downstream test turbines, and we assessed the impact of various offsets on those turbines. In addition to the
wind turbines, the site includes two ground-based lidars. Finally, prior to the experiment, we equipped several of the turbines
(T4,T5,T6,T3) with nacelle-mounted lidars.
Data from the turbines are reported in 10-minute averages and there was no possibility to inspect higher-frequency data.
This is different than the one-minute averages used in Fleming et al. (2020), for example. While 10-minute average data is
conventional for most supervisory control and data acquisition (SCADA) analysis in wind energy, we believe it can be a
challenge for wind farm controls analysis, which is highly sensitive to wind direction.

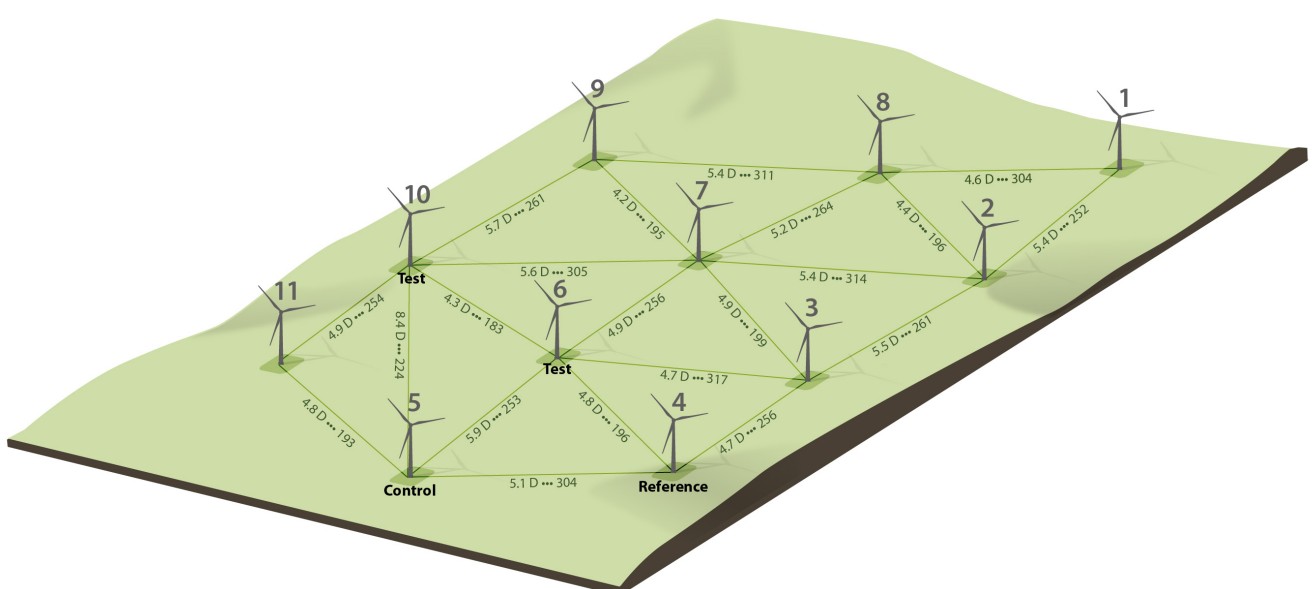

**Figure 1.** Layout, including interturbine spacings in terms of turbine rotor diameters (*D*), of the wind farm used in this experiment. The
figure also indicates the wind directions by which one turbine wakes another. See Fig. 2 for direction conventions.



A first step in preparing for the campaign was constructing the FLORIS model of the test site. An example simulation from this model is shown in Fig. 3. Details of the FLORIS wake model used are provided in the appendix.

With the first initial comparisons of the FLORIS model to the SCADA data collected before the start of the campaign, we observed that the absolute wind directions (those giving the compass direction of the wind, and not the vane-measured relative angle to the wind turbine nacelle) were approximately +9 degrees offset from FLORIS' expectations. We define a positive yaw offset as one in which the turbine is rotated counterclockwise of the incoming wind direction (as illustrated in Fig. 2).

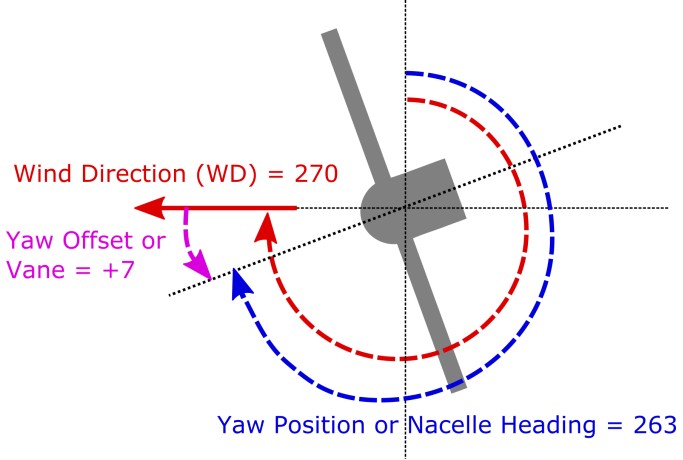

**Figure 2.** Illustration of conventions for the name and sign of certain direction signals. The figure illustrates what is meant by a +7-deg yaw offset.

Comparison with the two ground-based lidars revealed a similar discrepancy. Because four of the wind turbines had nacelle-mounted lidars installed, we were further able to assess that the relative yaw misalignment (i.e,. relative error between the turbine yaw angle and measured wind direction) of each of those turbines appeared to be close to 7 deg. We then assigned the residual error over 7 deg to the nacelle position encoders. Fig. 2 illustrates these direction signals and the sign and naming conventions used in this article. For the seven other turbines, we assumed that the error could be similarly divided between the vane and nacelle position error. These results are summarized in Table 1.

With these findings, we corrected the raw nacelle position and wind vane data using the offset values in Table 1. The assumed offsets imply that the normal condition of the wind farm is for all the turbines to have an approximately 7-deg offset. This update is incorporated into the FLORIS model of the wind farm by correcting the raw values. Fig. 3 shows an illustration of the finished FLORIS model of the site assuming these offset values.

Note that the offsets are positive angles, which in our convention implies a counterclockwise rotation, which is the direction we use for wake steering. Accounting for this issue when resimulating the data in FLORIS should mean the SCADA data and model can be directly compared. However, it is important to note that this default offset changes the nature of the test from a comparison of a yawed to an aligned wind turbine, to a comparison of a smaller and larger offset.



**Table 1.** Summary of direction offsets with respect to the ground and nacelle-mounted lidars.

| Turbine | WD Offset Ground Lidar | Nac-Lidar Measured Vane Offset | Assumed Vane Offset | Implied Nacelle Offset |
|---|---|---|---|---|
| 1 | 9.3 | | 7 | 2.3 |
| 2 | 13.1 | | 7 | 6.1 |
| 3 | 6.3 | 6.4 | | -0.1 |
| 4 | 7.3 | 6.8 | | 0.5 |
| 5 | 9.3 | 7.2 | | 2.1 |
| 6 | 9.2 | 7.1 | | 2.2 |
| 7 | 6.2 | | 7 | -0.8 |
| 8 | 6.0 | | 7 | -1.0 |
| 9 | 10.4 | | 7 | 3.4 |
| 10 | 14.1 | | 7 | 7.1 |
| 11 | 7.3 | | 7 | 0.3 |

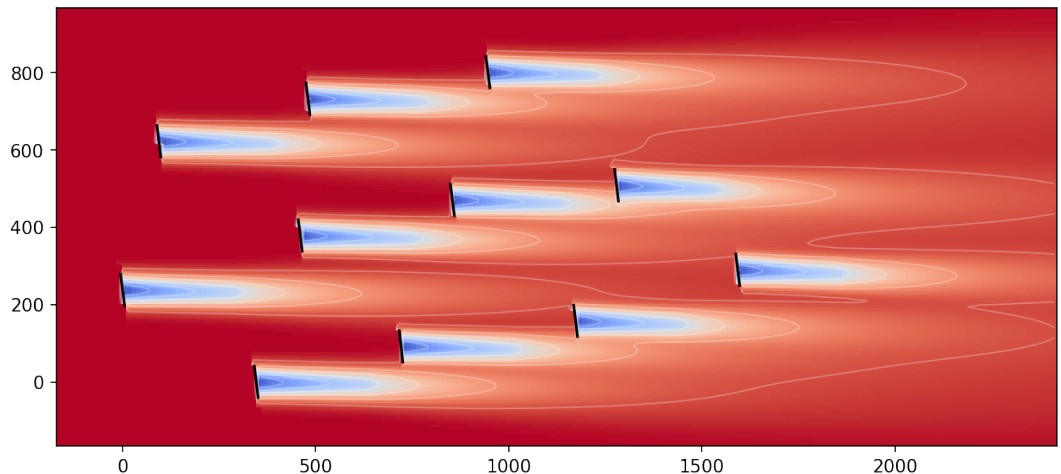

**Figure 3.** FLORIS model of the test site assuming the normal offsets provided in the middle columns of Table 1 for the case of winds from 270 deg.





## 3 Campaign Phases

Following the discovery of the systemic small positive offsets, we decided to divide the static test into three phases. In all phases, a target offset would be sent to T5, the controlled wind turbine. In the first phase, we applied an offset of -7 deg (a clockwise rotation) to confirm the lidar-measured offset by seeking to observe an increase in power of the controlled wind turbine when the offset is applied. Then, we applied a +10- and +20- (counterclockwise) offset in Phases 2 and 3.

The first phase -7 offset target test was run from February 18, 2020, through March 29, 2020. The second phase test targeting 100 +10-deg offset ran from April 7, 2020, through June 23, 2020. Finally, the +20-offset period ran from June 25, 2020, through November 2, 2020. Note that for this site, which is located in the Northern Hemisphere, the winds are most prevalent in the winter and least in the summer, so to some extent these periods are a bit unlucky in producing annual wind speeds that are lower than average.

In all phases, the controller is toggled on and off hourly to generate a data set with an equal amount of baseline (no applied 105 offset) versus controlled (offset applied cases), with a similar distribution of observed wind speeds and directions.

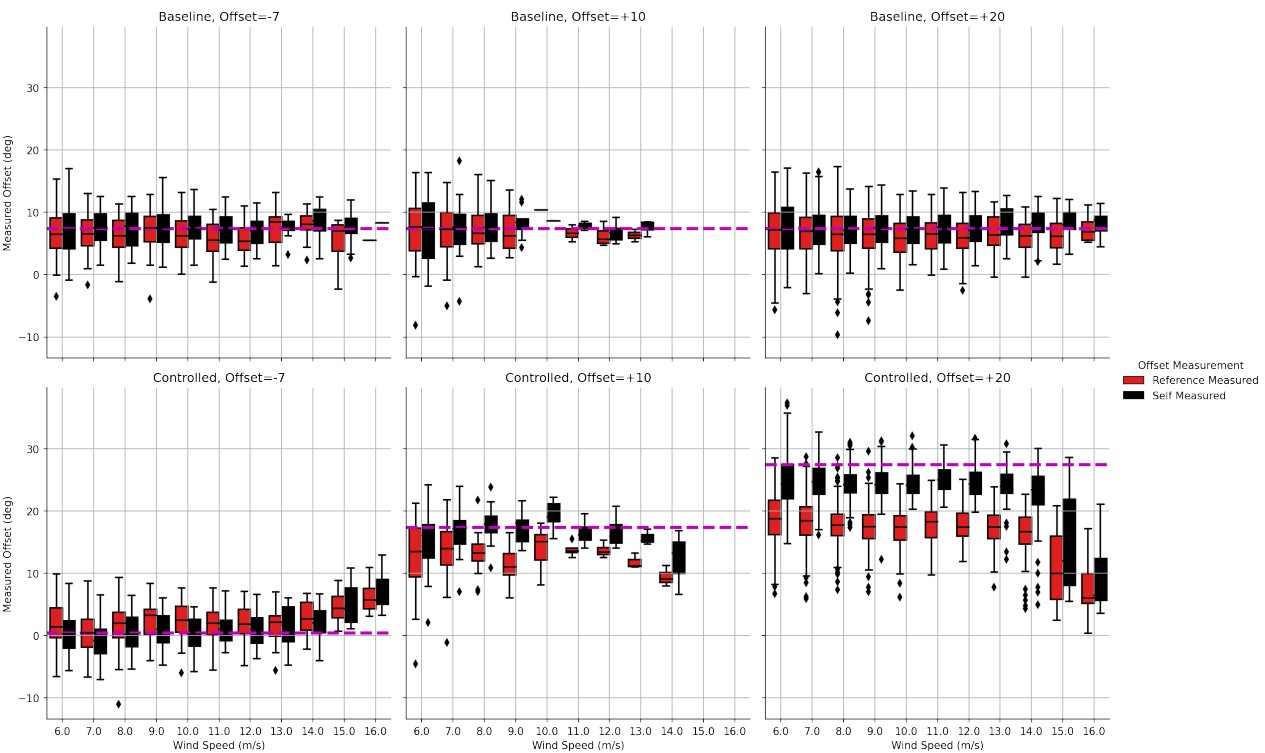

**Figure 4.** Boxplots, binned by wind speed, of the offset of the controlled wind turbine, as measured by its own vane (black) or in comparison to a reference wind direction (red). The targeted offset is indicated in each case with a magenta dashed line.





| Offset Target | Self Measured | Reference Measured | Difference From Baseline | Duration |
|---|---|---|---|---|
| 0 (Baseline) | 7.1 | 7.4 | (-) | (All cases) |
| -7 | 0.35 | 2.0 | -5.4 | Feb. 18–Mar. 29, 2020 |
| +10 | 17.1 | 13.6 | +6.2 | Apr. 7–Jun. 23, 2020 |
| +20 | 24.7 | 18.5 | + 11.1 | Jun. 25–Nov. 2, 2020 |

**Table 2.** Summary of self- and reference-measured mean offsets by target.

In Fig. 4, the yaw offsets of the controlled turbine (T5) are shown. The offsets measured both by the wind turbine's vane (black) and then computed by comparing its yaw position to the wind direction measured by the reference T4 (red) are shown for the three periods. The top row represents the baseline conditions when no offset is applied and so the wind turbine is at +7 deg measured either by itself or by the reference. In the controlled cases, there are a few details to note. First, the offset is limited to wind speeds of 14 m/s and below. Second, there is a certain amount of undershoot in the self- and reference-measured offsets. The average offsets are summarized in Table A1. The gap between self- and reference-measured offset grows with the size of the target offset and most likely reflects a need to apply corrections to the vane signal when operating in yaw (although it could also be partially attributable to noise issues). Note that when simulating the data in FLORIS, we used the reference-measured offsets.

## 4 Results

In this study, we first cleaned and filtered the collected data. The data are limited to directions of interest and wind speeds are limited to 14 m/s and below, as shown in Fig. 4, and that above 14 m/s the offset is turning off. Next, we removed any data that are flagged as curtailed or not fully operational for any of the relevant wind turbines. The data were then further filtered to remove power production anomalies using the filtering toolkit in NREL's OpenOA software (Perr-Sauer et al., 2021). Specifically, the power production data for each wind turbine was divided into 50-kW bins. Within each power bin, samples for which the nacelle-measured wind speed is more than 1 standard deviation from the median wind speed are treated as outliers and removed. Finally, the wind vane and nacelle measurements for all the wind turbines were corrected according to Table 1.

Next, we resimulated all the data points using FLORIS. The wind speed and direction are provided by the reference turbine (T4) (and not the lidars to maximize data availability, noting that the turbine's measurements are calibrated against the lidars).

### 4.1 Impact on the controlled wind turbine

We then assessed the impact of the three target offsets on the power production of the controlled wind turbine. We computed the power ratio of T5 over T4, binned by wind speed, over the range of directions for which both turbines are operating in freestream. The resulting power ratios for the baseline and controlled operation for the three target offsets are shown in Fig. 5.





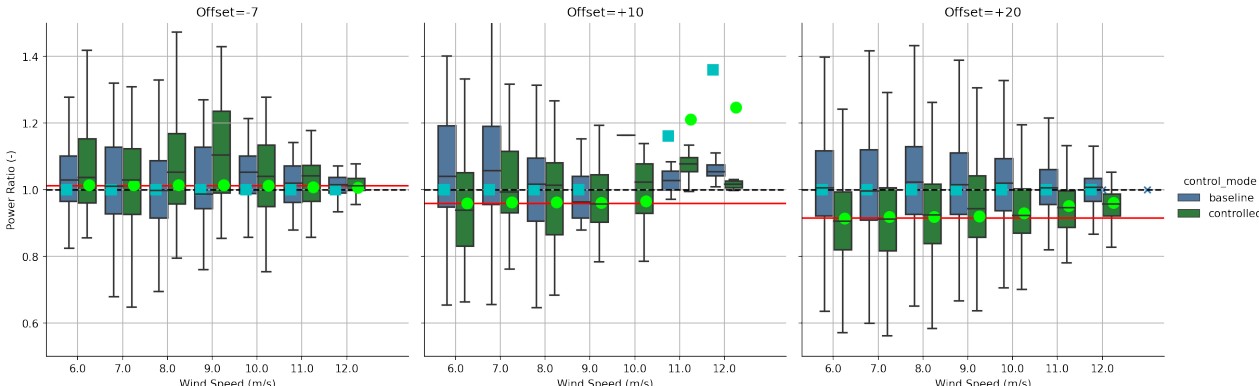

**Figure 5.** A comparison of power ratios of the controlled wind turbine (T5) with respect to the reference turbine (T4) for different target offsets. The power ratios are binned by wind speed and the results for baseline (blue) and controlled operation are compared. The circles indicate the expectations from the FLORIS model, whereas the red horizontal line is assuming a fixed cosine-squared loss.

Fig. 5 compares the power ratios of the collected field results with those predicted by the FLORIS model. Note that FLORIS models the loss caused by yaw as a change to the effective wind speed used in calculating power. Below rated when the wind turbine is following the optimal power production for wind speed, this loss model is equivalent to cosine-squared power loss per change in yaw angle. However, this loss is reduced in the transition to rated and well above rated the cosine exponent goes to zero.

The model achieved a good match between the measured and predicted losses for the +20-deg offset phase. The other phases agree regarding direction, with the power ratios increasing for a -7-target offset (confirming again that the wind turbine was initially offset positively). In the case of the +10 offset, the agreement is good for lower wind speeds, wherein suitable data were collected, but above 8 m/s, the amount of data collected is too little, and both the field and FLORIS results indicate that the underlying data are not sufficient for realistic convergence. The function by which turbines lose power with increasing

yaw angle is likely to be wind-turbine- and turbine-controller-dependent and is itself a subject of research (see, for example, Howland et al. (2020)). Finally, the decreasing size of the boxes with increasing wind speed is mostly likely related to the steadier power level in high wind speeds.

### 4.2   Impact on downstream wind turbines

We now consider the impact of fixed offsets and wake steering on the downstream wind turbines. As shown in Fig. 1, we use

T6 (5.9 *D*) and T10 (8.4 *D*) as the downstream turbines. T3 would have been an additional candidate; however, in those wind directions the change in wake direction from wake steering can affect the reference wind turbine power production.

Going forward, we limit comparisons to the third phase experiment, wherein a +20-deg offset is targeted. However, it is worth reiterating that according to Table A1 the baseline offset is 7.4 deg, and the controlled offset is 18.5 deg, making this a comparison between a small offset angle and a larger one, not an offset versus an aligned condition, which we originally





planned to do. Still, by applying the measured offsets and not the target offsets in FLORIS, we can make a fair comparison to the model, and account for undershoot, as shown in Fig. 4.

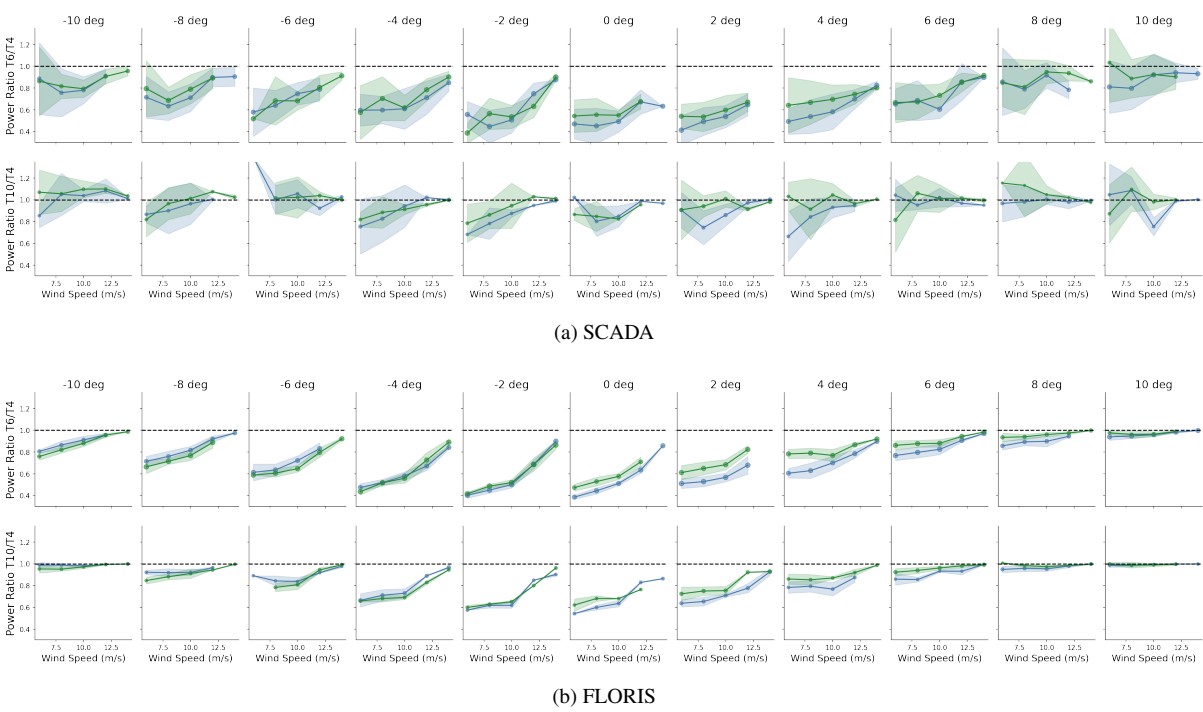

**Figure 6.** Comparison of power ratios for T6 and T10, with respect to reference T4, for both SCADA data and FLORIS resimulations. The size of the circles indicates the number of points in a given bin and the shaded area indicates a 95% confidence interval of the mean. Note that wind direction has been shifted in these plots so 0 deg is the angle when the downstream wind turbine is fully waked. To save space, the legend is not shown, but as in Fig. 5, green represents the controlled case and blue is the baseline.

Fig. 6 compares the power ratios, which are binned by wind speed and direction. The overall trends observed in the data show consistent improvement across the range of direction where wake steering is expected to be beneficial (0 deg and above in the shifted coordinates of the plot).

To facilitate a comparison between SCADA data and model predictions, it is useful to aggregate across wind speed bins to combine the data into a single result per wind direction. As explained in Fleming et al. (2019), our preferred method is called the energy ratio, which is a ratio of the weighted binned mean power of the test and reference wind turbines (and not a mean power ratio), wherein the bins are wind speed and weights are the frequency of occurrence of each wind speed bin. The baseline and controlled data sets are weighted by the same distribution (to control for any disparity in distribution) and any points in

the wind speed bins without at least one corresponding point in the alternative control bin are removed. Uncertainty bands are computed via bootstrapping to represent 95% confidence. The source code implementation of the method and example implementations are included with FLORIS. The resulting energy rations are shown in Fig.7.



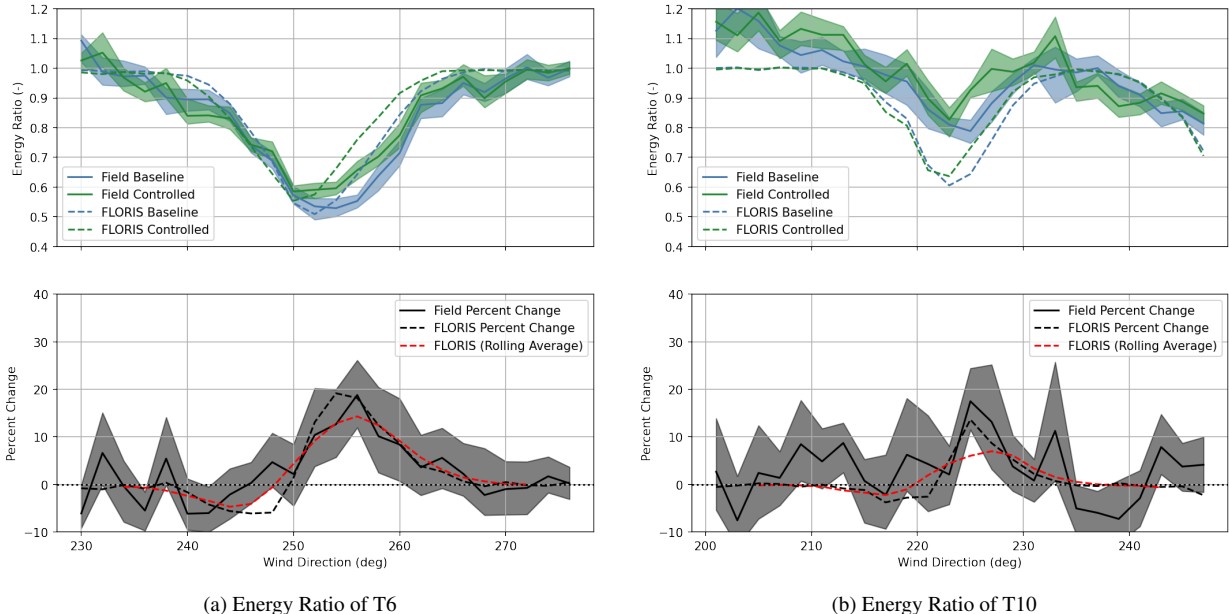

(a) Energy Ratio of T6                        (b) Energy Ratio of T10

**Figure 7.** Comparing the energy ratios and percent change in energy ratio for test turbines T6 and T10, with respect to T4. The FLORIS values are indicated as dashed lines. In the percent change plot, the percent change in FLORIS is indicated in black, and the rolling window average is indicated in red.

The energy ratios for T6 and T10 with respect to T4 are shown in Fig. 7. The upper plots show the energy ratios themselves (with values from FLORIS indicated by dashed lines) and the lower plots percent change. The values from FLORIS appear to
be fit well. One discrepancy is that the ratio value of T10 does not center around 1.0 to the left of the wake zone (directions less than 215 deg). We referred to computational fluid dynamics analysis of the site to confirm that this is an expected consequence of local terrain.

One surprise is that although the gains made on the right half of the wake profile (where the wake is steered away from the downstream wind turbine) are approximately in line with FLORIS' expectations, the losses on the left side (where the wake
is steered toward the downstream wind turbine) are less than expected. Part of this we believe is attributable to the 10-min averages that most likely blur the close peaks in gain and loss near the energy ratio nadir, and for this reason the red-dashed 10-deg rolling-average version of the FLORIS result is shown to indicate this type of effect. However, even still there is a very consistent asymmetry in wake steering where even accounting for blurring, the gains of correct steering, are more than the losses from wrong-way steering.

In the recent paper by King et al. (2020), the GCH model of wake steering is introduced to better capture the effects of wake steering. One second-order effect included in the model predicts that wake steering reduces the total wake deficit by inducing yaw-added recovery. This recovery is driven by the counter-rotating vortices generated in wake steering and is in addition to the reduced wake deficit caused by the reduction in thrust from yawing. It is also different from this reduction in initial deficit



in that the effect increases with distance travelled. This property implies that in a constant offset experiment like this one, the
gains from "right-way" wake steering will exceed the losses observed in "wrong-way" wake steering. It is important to note
that these gains and losses are in terms of the downstream wind turbine only and not combined upstream-downstream power.
This asymmetry is clear in these results and the measured losses from wrong-way steering are less than the amount already
reduced by incorporating the GCH model. Also, it can be observed that, taking into account the uncertainty, it at least appears
the loss is even less for the farther turbine (T10), in agreement with predictions of yaw-added recovery.

We believe the asymmetry in the impact of wake steering on power is an important result for two reasons. It provides
supporting evidence to the physical models proposed in Martínez-Tossas et al. (2019) and King et al. (2020). These can be
combined with wind tunnel studies such as Zong and Porté-Agel (2020); Campagnolo et al. (2020) to gain confidence in these
wake-steering models. First, using these models in the design of wind farm controllers produces different control strategies
from prior deflection models and can be critical in the design of control strategies for large arrays, because of the secondary
steering effects (see, for example, Zong and Porté-Agel (2021)). Second, the degree of power loss from wrong-way steering
affects the design of robust control strategies, which are intended to address wind direction measurement uncertainty as well
as the inability of the controller to track high-frequency wind direction variations because of slow yaw control dynamics (Rott
et al., 2018; Simley et al., 2019; Quick et al., 2020). The penalty paid for wrong-way steering directly affects the magnitude of
the yaw offsets applied for wind directions where wake steering is beneficial. Overestimating the power loss from wrong-way
steering could lead to overly conservative yaw offsets and, consequently, less energy gain from wake steering than could have
been achieved. Therefore, accurate predictions of power loss are important to maximizing the effectiveness of wake steering.
Further, when evaluating the potential of wake steering for commercial projects, assuming wind direction uncertainty, the
degree of asymmetry in the change in power from wake steering will influence the expected annual energy production.

Fig. 7 presents the change in energy ratio combining all available data. However, as discussed in Fleming et al. (2020), for
example, it can be useful to divide the data and analysis to see how things are similar or disparate (e.g., in stable versus unstable
atmospheric conditions). Analysis in this view can lead to control designs tailored toward each condition (see for example Ruisi
and Bossanyi (2019).

In this work, we notice the greatest difference in effect is when we divide the data in half according to the standard deviation
in wind direction measured by the reference wind turbine. These divided results for both downstream turbines are shown in
Fig. 8. This division shows a very clear change in the energy ratios as well as the apparent performance of wake steering.
Unfortunately, there is an ambiguity to the interpretation: does separating the data by wind direction deviation select from
identical underlying conditions those times where the signal to noise is highest (because the fewest wind directions are mixed
together in the averages), or is it filtering by atmospheric conditions that have higher and lower variability (stable versus neutral,
for example), as well as other correlated properties (such as turbulence level)? Perhaps both?

Finally, in Figures 9 we show the energy ratios, and percent change, for the combined powers of upstream and downstream
turbines. With reference to the FLORIS model, you can see that the controlled offset (18.5 degrees) would for most wind
directions be expected to produce an overall loss in combined power relative to the baseline offset (7 degrees). An overall
improvement is only expected for a small range of wind directions for the T5/T6 pair, and no discernible improvement is



(a) Results for T6

(b) Results for T10

**Figure 8.** Energy ratio and change in energy ratio, as in Fig. 7; however, divided into two subsets according to the wind direction variability measured by the reference wind turbine. As estimated by FLORIS, increasing the baseline offset of 7 deg to the controlled offset of 18.5 deg is only beneficial for a small sector of wind directions.





expected for the T5/T10 pair (the best that FLORIS predicts in this case is that we can 'break even' for a small direction range).
The results shown here are consistent with these expectations from FLORIS.

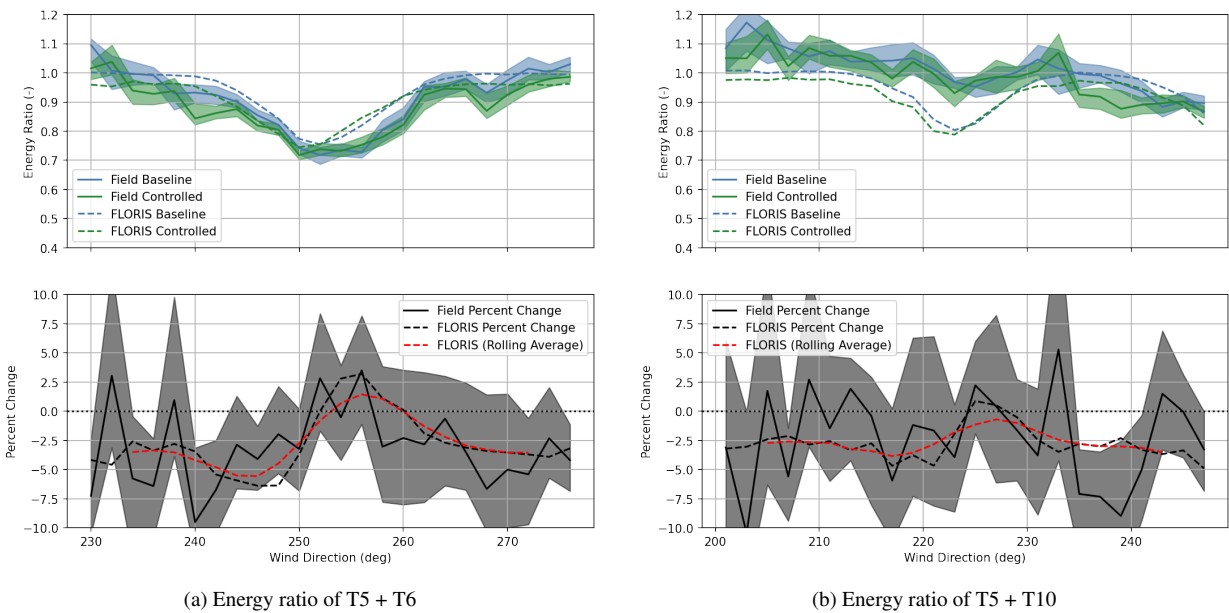

(a) Energy ratio of T5 + T6                    (b) Energy ratio of T5 + T10

**Figure 9.** Energy ratio and percent change for combined upstream and downstream wind turbine power.

## 5 Conclusions

This article presents results from a wake-steering study using fixed yaw offset targets, instead of a lookup table of optimal yaw offsets as a function of wind speed and direction. The results allowed us to investigate model predictions in conditions that would normally be avoided, such as intentional wrong-way steering. Comparing the model and measured results shows
that the gains from right-way steering were in line with expectations, whereas the losses from wrong-way steering were less. The lower-than-expected losses from wrong-way steering are particularly promising because the GCH model of wake steering to which the results are compared already predicts a decrease in losses compared to the standard Gaussian wake model as a consequence of the "yaw-added recovery" effect.

We believe this article adds to the growing literature that validates wake steering at a range of scales and fidelities. Com-
225 parisons between model predictions and SCADA results were generally positive within the limits of the data uncertainty. The results show that engineering wake-steering models for small numbers of wind turbines and when wind-direction-based offset strategies and associated control implementation difficulties are removed, are reasonably accurate.

For wind farm control to be realized in practice in industry, validation campaigns which increase confidence in the engineering models used in the design of wind farm controllers are critical.Boccolini et al. (2021). We believe this campaign contributes





importantly to that need. However, the campaign, like previous campaigns, is of only a limited number of turbines. The performance of wake steering in arrays of turbines larger than 10 would validate additional important effects, such as blockage, secondary steering, wind farm boundary layers, and deep-array effects, all of which may pose problems that do not manifest themselves at the scale of a few turbines. Future research campaigns should address these issues through implementations on larger numbers of turbines.

Finally, design and analysis of controllers for achieving the maximum possible performance, given practical issues such as modeling error, wind direction variability, and yaw activity limitations, as well as opportunities to match control strategies to specific atmospheric conditions, are recommended for future research.

*Code availability.* FLORIS is available for download at github.com/NREL/floris



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

*Author contributions.*  PF, MS, TY, and ML designed the experiment. PF, MS, TY, ML, JK, ES and BD contributed to analysis and interpretation of the data. PF prepared the manuscript with contribution from all co-authors.

*Competing interests.*  The authors declare that they have no conflict of interest.





| Parameter | Value |
|---|---|
| Turbulence Intensity | 0.09 |
| Velocity Model | gauss_legacy |
| ka | 0.38 |
| kb | 0.004 |
| calculate_VW_velocities | true |
| use_yaw_added_recovery | true |
| Deflection Model | gauss |
| dm | 1.0 |
| use_secondary_steering | true |
| Turbulence Model | crespo_hernandez |
| initial | 0.5 |
| constant | 0.9 |
| ai | 0.75 |
| downstram | -0.325 |

**Table A1.** Summary of self- and reference-measured offsets by target.

## Appendix A: FLORIS Appendix

Details of wake model used in FLORIS simulation.