# Peer review of "Experimental results of wake steering using fixed angles"

_Wind Energy Science, 2021_

## Author Comment (AC1)

**Reviewer 1**

**General comments:**

This research article investigates wake steering with fixed yaw offset in a commercial wind farm. The use of fixed yaw offsets allows for comparison with wake model predictions, which is valuable for further confidence in the models and future implementation in optimized wake steering algorithms.

In this article power ratios from a commercial wind farm are compared to model predictions from the newest version of the FLORIS model, which generally shows very good agreement with the SCADA results assessed from the turbines. Although there still seem to be some uncertainties related to offsets in wind direction, the reasoning for the implemented correction seems logical and the presented results show good agreement with model predictions. The influence of second-order effects in the wake model predictions, such as the asymmetry in wake steering, secondary steering and the yaw added wake recovery due to Gauss-curl is discussed. Furthermore, the results are comprehensively discussed with results from simulations and wind tunnel experiments.

Some important details about the wind turbines and farm setup are sparsely described, which might be due to restrictions from the farm operator. If this is the case, it gives some limitations to the repeatability of the work. However, the presented work is assessed to be an important contribution on the way to implementing optimized wind farm control in practice. Overall, I recommend this article for publication. I have only some minor comments, most of which are insignificant technical corrections.

We thank the reviewer for this assessment, and will address the issue of wind turbine details in the response to specific comments.

**Specific comments:**

P3.L62. "Results show agreement between model predictions and measured results." This sentences sounds very conclusive to end the introduction. Maybe reformulate "This research article will investigate agreement between …"?

This sentence is revised to: *Finally, model predictions are compared to measured results to assess agreement.*

P3.L64. Test site overview: very good description of the turbine setup, but I could not find any information about the type and size of the wind turbines itself. I guess this information might underlie confidentiality agreement with the farm owner, but – if possible - some basic info about the approximate rotor diameter, rated power and thrust coefficient would be very useful for comparison with future datasets.

We can't share exact information for the reasons you guess, but we have added these sentences: *All turbines at the test wind farm are the same model and dimensions. Exact details are withheld, but the rated power is close to 2MW, the rotor diameter is in the range 80m to 90m and the hub height is between 60m and 70m.*

P3.L68. "… two ground-based lidars". Where are these located in/next to the farm? Could the locations be included in Figure 1?

We have added the following text: *One of the ground-based LiDARs was located approximately 2.5 rotor diameters south of T4. The second was located approximately 2.5 rotor diameters north of T5.*

P3.L71. "While 10-minute average data is conventional (…), we believe it can be a challenge…". I do strongly agree to that statement. Can you link this to the statement on P10.L170?

This is done, the new text reads: *Part of this we believe is attributable to the 10-min averages that most likely blur the close peaks in gain and loss near the energy ratio nadir, and is a reason we think 10-minute averages can present a challenge in this research.*

P4."Figure 2" It took med some time to figure out the different offsets you measured. Would it be possible to include "compass direction" "vane direction" and "real wind direction" in the figure?

The figure has been updated per recommendation

P5."Figure 3. FLORIS model…" Is it possible to include the turbine numbers also in this figure?

The figure has been updated to include turbine numbers

**Technical corrections:**

P1.L23. "… necessary that the models **are** computationally efficient."

Done

P2.L24. "... (FLORIS) software framework (Laboratory (2019)) ...". Looks like the automatic referencing interpreted "laboratory" as an authors last name. Change to "NREL" or similar

Adding curly brackets fixed the issue

P7.L111. "Table A1". I assume "Table 2" is the correct reference here.

Fixed

P8.L148. "Table A1". As above I assume "Table 2" is the correct reference here.

Fixed

P10.L171. "...energy ratio nadir". I am not sure if "nadir" is a typo or has a meaning here.

Changed to low point

P11.L209. "...correlated (such as turbulence level?) Perhaps both?". This sentence is somewhat informal. Could you reformulate? I do however agree with the interpretation of these effects.

For better clarity, this sentence is broken into several: *Unfortunately, there is an ambiguity to the interpretation: does separating the data by wind direction deviation select from identical underlying conditions those times where the signal to noise is highest (because the fewest wind directions are mixed together in the averages)?  Alternatively, is the division separating the data by atmospheric conditions that have higher and lower variability (stable versus neutral, for example), which then in turn separates the data by the correlated properties of turbulence level, meaning the change is physical and not simply a signal to noise issue in the analysis.  Or most likely, the separation is both a division by signal to noise, and a division correlated to atmospheric conditions.*

P13.L229. "...are critical.Boccolini et al. (2021)". Revise punctuation.

Fixed

P18. "Table A1. Summary of selv- and reference measured offsets by target". I assume the capture text is not correct as it is the same as for "Table 2".

Fixed

**Reviewer 2**

**General Comments**

The paper describes the results of an experimental campaign with fixed yaw offsets on a small wind farm. The results are used to investigate wake-steering asymmetry and validate model predictions of recent additions to the FLORIS modelling tool.

The experimental design seems well thought-out and the results appear to support the conclusions that are drawn. The findings are thoroughly discussed with respect to other work in the field. However, little information is provided on the wind farm that is studied.

In my opinion, the paper provides a valuable contribution the wind energy community. It would benefit from a little more information on the wind farm being measured and the inclusion some quantitative measure of the model prediction quality.

We thank the reviewer for these comments!

**Specific Comments**

- Could you provide more details of the wind farm? What is the size of the wind turbine used in the experiment?

We can't share exact information, but we have added these sentences: *All turbines at the test wind farm are the same model and dimensions. Exact details are withheld, but the rated power is close to 2MW, the rotor diameter is in the range 80m to 90m and the hub height is between 60m and 70m.*

- What is the variability on the measured wind vane offset? Is it valid to assume it to be constant?

We don't assume it to be constant.  Figure 5 provides an indication of the variability of wind vane offset for both the baseline and controlled conditions.  The boxes cover the lower to upper quartile of values, with the whiskers given the full range minus the "flier" points.

- Visually, the model seems to achieve a "good match" (135). What is considered "good"? Could you quantify the quality of the predictions in terms of percentage error, RMS, or a similar measure?

We have revised this sentence to be clearer on what is meant by good match to be following the trends seen in low and higher wind speeds.  The new text reads: *The model's predicted losses for the +20-deg phase agree well with the trends observed in the field, with a cosine-squared loss in the lower wind speeds, and decreasing losses as wind speeds increase.*

**Technical Corrections**

- Parenthesized citations should be adjusted to (Author, year) instead of (Author (year))

This is constructed by the LATEX class file, we will work with journal editors to correct these style issues in final copy

- Several figures have a font size that is too small to read, please increase size.

We've tried to correct this by moving the legend inside the plot for Fig 5, so that it and the figure can be larger, and elsewhere by describing the legend within the caption of the figure.

- 19 "... turbines. Wagenaar et al. (2012) Research" - adjust citation formatting

This is set by the provided LATEX class, we will work with the journal editors on final formatting.

- 24 "analytical, or statistical," - statistical is not equivalent to analytical, remove comma's

These are removed, thank you

- 24 "(Laboratory (2019))" - adjust reference to NREL

Fixed

- 39 "(Churchfield et al. (2014).)" - comma after closing parenthesis

Fixed

- 61 "Results show agreement between model predictions and measured results." - this statement does not belong in the introduction.

This sentence is revised to: *Finally, model predictions are compared to measured results to assess agreement.*

- 117 "and below, as shown in Fig. 4, and that above 14 m/s the offset is turning off." - adjust sentence structure

This is now split into 2 sentences for clarity: *The data are limited to directions of interest and wind speeds are limited to 14 m/s and below.  The data are limited this way because  as shown in Fig.~\ref{fig:offsets}, above 14 m/s the offset is turning off*.

- 162 - "energy rations" - should be "energy ratios"?

Fixed

- 173 "... the gains of correct steering, are more ..."  - redundant comma

Fixed

- 229 "critical.Boccolini et al. (2021)" - adjust citation formatting

Parenthesis added

- Figure 3 - it would be good to have the turbines be numbered

Numbers now added

- Figure 6 - a legend would be of value for clarity, could you consider another way to save space?

If it is acceptable to the reviewer, we struggled to include a legend into this figure as it is hard to fit this figure, which we presents the data in valuable and transparent way. We note that the figure is fully explained in the caption and the colors are consistent with all other figures referring to baseline and controlled.

- Figure 4 - might add the number of the controlled turbine here as well

This is added

- Table 2 - move caption above table

This is done